# Cardio-Respiratory Monitoring in Archery Using a Smart Textile Based on Flexible Fiber Bragg Grating Sensors

**DOI:** 10.3390/s19163581

**Published:** 2019-08-17

**Authors:** Daniela Lo Presti, Chiara Romano, Carlo Massaroni, Jessica D’Abbraccio, Luca Massari, Michele Arturo Caponero, Calogero Maria Oddo, Domenico Formica, Emiliano Schena

**Affiliations:** 1Unit of Measurement and Biomedical Instrumentation, Università Campus Bio-Medico di Roma, Via Alvaro del Portillo, 00128 Rome, Italy; 2Neuro-Robotic Touch Laboratory, Biorobotics Institute, Sant’Anna School of Advanced Studies, 56025 Pisa, Italy; 3Department of Linguistics and Comparative Cultural Studies, Ca’ Foscari University of Venice, 30123 Venice, Italy; 4Photonics Micro- and Nanostructures Laboratory, ENEA Research Center of Frascati, 00044 Rome, Italy; 5Unit of Neurophysiology and Neuroengineering of Human-Technology Interaction, Università Campus Bio-Medico di Roma, Via Alvaro del Portillo, 00128 Rome, Italy

**Keywords:** fiber Bragg gratings, smart textiles, wearable systems, cardiac monitoring, respiratory monitoring, precision sports, archery

## Abstract

In precision sports, the control of breathing and heart rate is crucial to help the body to remain stable in the shooting position. To improve stability, archers try to adopt similar breathing patterns and to have a low heartbeat during each shot. We proposed an easy-to-use and unobtrusive smart textile (ST) which is able to detect chest wall excursions due to breathing and heart beating. The sensing part is based on two FBGs housed into a soft polymer matrix to optimize the adherence to the chest wall and the system robustness. The ST was assessed on volunteers to figure out its performance in the estimation of respiratory frequency (f_R_) and heart rate (HR). Then, the system was tested on two archers during four shooting sessions. This is the first study to monitor cardio-respiratory activity on archers during shooting. The good performance of the ST is supported by the low mean absolute percentage error for f_R_ and HR estimation (≤1.97% and ≤5.74%, respectively), calculated with respect to reference signals (flow sensor for f_R_, photopletismography sensor for HR). Moreover, results showed the capability of the ST to estimate f_R_ and HR during different phases of shooting action. The promising results motivate future investigations to speculate about the influence of f_R_ and HR on archers’ performance.

## 1. Introduction

Archery is a precision sport which requires consistency and stability of movements [1]. A mismatch of physical, physiological, and psychophysical factors can influence athletic performance and deteriorate archers’ accuracy and precision [2,3,4,5].

In precision sports, both breathing and heart rate (HR) influence the athlete’s performance [5,6,7]. The control of such physiological activities facilitates the performance of repetitive shots in the same, stable posture [8]. Lakie et al. demonstrated that high values of HR can cause sway movements, tremor, and shaking of the body when aiming at the target [8]. Mohammed et al. showed that HR variations affect breathing capacity, thus inconsistent breathing patterns can also impose a negative effect on the heart rate, and in general on the athlete’s performance [9].

In archery, the shooting action can be separated into three main phases: set-up, aiming, and release. During the set-up phase, shoulders are brought in line with the target, hips are rotated forward, and the hand position on the bow grip is set-up. During the aiming phase, the focus is completely diverted to the target and the alignment of bow-sight-target is performed. The shooting of the arrow takes place in the release phase. It is highly recommended that archers start to inhale during the set-up phase of shooting, and to either exhale or hold their breath during the aiming phase. This allows for reducing the level of body rigidity and, simultaneously, preparing the body for the release phase [5,10].

As a result of training, master marksmen know exactly how to time the release of the arrow with the pattern of their breathing and cardiac cycle, thus minimizing the body jerk caused by the breathing process and the heart contraction [11]. Expert athletes experience similar breathing patterns and low HR values during each shot. On the contrary, unskilled archers hold their breath for longer during the aiming phase. The forced breathing process often leads to sway movements due to muscular contraction and an increase of ventricular depolarization, compromising body stability during the aiming phase [12,13,14]. Thus, comprehensive monitoring of breathing and heart beating during the shooting phases can improve the scheduling of exercises and optimize the training strategies [9].

Breathing and cardiac activities can be monitored by several solutions [15,16]. Among them, smart textiles based on fiber Bragg grating sensors (FBGs) have gained broad interest to monitor the mentioned vital signs in an unobtrusive and comfortable way [17,18]. FBGs can be easily incorporated into textiles thanks to their small size and lightweight. In addition, the high sensitivity and adequate bandwidth make these sensors an optimal solution for such an application. Some potential drawbacks may include the difficulty to handle bare optical fibers, their tiny resistance to mechanical stress, and the requirement to be connected to an optical spectrum interrogator. The encapsulation of FBGs into soft and flexible polymer matrices allows for mitigating the mentioned limits [19,20,21]. This solution makes easy to handle the fiber and improve the contact compliance with the body, leading to more accurate measurement of parameters from the chest surface movements. At the same time, recent progress towards the development of miniature FBG interrogation systems may broaden the application of FBGs for continuous and remote monitoring [22].

The aim of the present study is the feasibility assessment of a custom ST based on flexible FBG sensors for cardio-respiratory monitoring in archery. The proposed system was assessed on volunteers during quiet breathing and apnea, as well as on archers during shooting sessions.

## 2. Principle of Work of the Custom Smart Textile

Two flexible sensors were used to develop a ST consisting of two elastic bands (600 mm × 40 mm × 2.1 mm, 10 kgf of maximum load and 100% of polyamide) worn around the thorax and the abdomen, respectively. The fit of each band was adjusted by VELCRO^®^ fastener to put the two FBGs in contact to the chest in correspondence of the xiphoid process and the umbilicus. The use of anatomical landmarks allows the FBGs to be worn on the same measurement points. Each sensor consists of an FBG (Bragg wavelengths λ_B_ of 1541 nm and 1545 nm for the band around the thorax and the abdomen, grating length of 10 mm and reflectivity of 90%; At Grating Technologies), previously housed in a flexible polymer packaging (90 mm × 24 mm × 1 mm) made of Dragon Skin^®^ 20 (Smooth-On, Inc. USA) as shown in Figure 1. A detailed description of the manufacturing process and of the sensors’ metrological properties are reported in [19].

FBGs work as stop band filters of wavelength because they back-reflect a small portion of light traveling along the fiber at a specific wavelength (i.e., λ_B_). The FBG working principle is well described by the following equation:(1)$$\lambda_{B}=2\cdot \eta_{eff}\cdot \Lambda$$
where *η_eff_*_,_ is the effective refractive index of the fiber core and Λ, the grating period. The dependence of *η_eff_* and Λ from temperature and strain makes FBGs an optimal solution for the development of measurement systems able to sense these two parameters [23].

Regarding the application of interest, breathing and heart beating cause periodic displacements of the chest and, in turn, stretch the flexible sensors embedded in the ST, as schematically reported in Figure 1.

## 3. Feasibility Assessment of the Smart Textile on Healthy Volunteers

### 3.1. Population and Experimental Protocol

The ST for cardio-respiratory monitoring was assessed on nine healthy volunteers (four males and five females) whose age and anthropometric characteristics are shown in Table 1.

Each participant was asked to perform a protocol consisting of three main phases: *i)* a short apnea useful to synchronize the reference instruments (i.e., flowmeter for f_R_, photopletismography sensor -PPG- for HR) and the FBG outputs; *ii)* 16 quiet breaths; *iii)* a final apnea as long as each volunteer can. The study was approved by the local ethical committee (protocol number 27/18).

### 3.2. Experimental Set-Up

The vital signs under investigation were monitored by the ST. The flexible sensors were positioned corresponding to the xiphoid process and the umbilicus (see Figure 2). Each FBG embedded into ST (FBG^T^ for the band around the thorax and FBG^A^ for the band around the abdomen, *a* box in Figure 2) was connected to an optical spectrum interrogator (si425, Micro Optics Inc., *b* box) which worked at the sampling frequency of 250 Hz. The reference signal for the respiratory activity was collected by a commercial flow sensor (SpiroQuant P, EnviteC, Alter Holzhafen, Wismar, Germany, *c* box in Figure 2) connected to a differential pressure sensor (163PC01D75, Honeywell, Minneapolis, MN, USA). The output of reference system used for the respiratory monitoring was collected by using a DAQ (NI USB-6009, National Instrument, Rockville, MD, USA, *d* box in Figure 2) and a custom virtual instrument developed in LabVIEW® environment, at the sampling frequency of 250 Hz. The reference system for the cardiac monitoring was a photopletismography sensor (*e* box in Figure 2), placed on the index fingertip of the left hand, as in [24]. The PPG sensor was input into two other analogue ports of the same DAQ used for the respiratory monitoring and collected at the sampling frequency of 250 Hz.

### 3.3. Data Analysis and Results

This paragraph will be grouped into two subsections according to the stages of the protocol (i.e., quiet breathing and apnea). Each subsection describes the data analysis performed to estimate f_R_ (during quiet breathing) and HR (during apnea), and the obtained results.

#### 3.3.1. Respiratory Frequency Estimation During Quiet Breathing

For each volunteer, data were processed following four main steps: *i)* the outputs of the FBGs and the flow sensor were synchronized by selecting the first minimum points (i.e., starting points) after the starting apnea (see Figure 3); *ii)* the quiet breathing stage was selected by cutting all the synchronized signals from the mentioned starting points and considering all the 16 breaths performed during the protocol (see Figure 3); *iii)* a filtering stage consisting of a second order pass-band filter (lower cut-off frequency of 0.05 Hz and higher cut-off frequency of 0.5 Hz) was applied on both FBGs and flow sensor outputs; *iv)* a custom algorithm was used to select the maximum peaks of each signal (see Figure 4).

The respiratory periods (i.e., T_R_^A^ and T_R_^T^ for the band around the abdomen and the thorax, respectively, and T_R_^FLOW^ for the flow sensor) were calculated as the time interval between two consecutive maximum peaks. Then, the f_R_ values estimated by both the FBGs (i.e., f_R_^A^ and f_R_^T^) and the flow sensor (i.e., f_R_^FLOW^) were calculated as the ratio between 60 and the related respiratory periods in order to express f_R_ in acts per minute (i.e., apm).

The ST was assessed in terms of both breath-by-breath and mean f_R_ values. The Bland-Altman analysis was performed to describe the agreement between f_R_ values estimated by the proposed system and the reference one [25]. This analysis considered all the f_R_ values of the enrolled volunteers (i.e., a total of 135 f_R_ values for each Bland-Altman analyses) for the calculation of the mean of difference (MOD) and the limits of agreement (LOA^+^ = MOD + 1.96·SD and LOA^−^ = MOD − 1.96·SD, where SD is the standard deviation of the differences between the data collected by the proposed system and the reference one). The mentioned analysis was performed considering data provided by each flexible sensor. The mean absolute percentage error in the f_R_ estimation (i.e., MAPE^f^^R^) was used to compare mean f_R_ values and was calculated as:(2)$$MAPE^{f_{R}}=\frac{1}{n}\cdot \sum \frac{\left|f_{R}{}^{smart\_textile}-f_{R}{}^{reference}\right|}{f_{R}{}^{reference}}\cdot 100$$
where $f_{R}{}^{smart\_textile}$ and $f_{R}{}^{reference}$ denote the values of the f_R_ obtained by the proposed system and the reference one, respectively.

Results of the breath-by-breath analysis are shown in Figure 5 and in Table 2. The good agreement between the f_R_ values measured by the proposed system and the reference one is confirmed by the high value of the correlation coefficient (R^2^) for both FBG^T^ and FBG^A^ and by the low value of both MOD and MAPE^f^^R^.

#### 3.3.2. Heart Rate Estimation During the Apnea.

The HR estimation during the apnea was performed according to the following four main phases: *i)* for each trial, the first minimum points after the holding of breath were selected on the FBGs output to define the starting point of the apnea stage for the FBGs and the PPG sensors; *ii)* the same time interval (i.e., 10 s) was chosen to estimate HR of all the volunteers during the apnea (see Figure 6); *iii)* a fourth-order Butterworth pass-band filter with a lower cut-off frequency of 0.6 Hz and a higher cut-off frequency of 20 Hz was applied on the signals. This band of frequency was chosen according to the frequency components of vibrations induced on the chest wall by the blood flow ejection into the vascular bed [26]; *iv)* a custom algorithm was used to select minimum peaks (blue markers in Figure 7) on each filtered FBG signal. The beat-by-beat cardiac period (T_C_) from the FBG outputs was calculated considering the minimum peaks, as the time elapsed between two consecutive minimum peaks (T_C_^T^ and T_C_^A^ from the band around the thorax and the abdomen, respectively). Minimum peaks were chosen because they are easier to detect on the filtered FBG signals, automatically. Beat-by-beat cardiac periods were calculated considering the time interval between two consecutive maximum peaks on the filtered PPG signal (T_C_^PPG^). The HR values (i.e., HR^A^, HR^T^, and HR^PPG^) were calculated as the ratio between 60 and T_C_^T^, T_C_^A^, and T_C_^PPG^, in this way HR is expressed in bpm.

The ST capability of monitoring HR was assessed in terms of beat-by-beat and mean HR values. The Bland-Altman analysis was performed by comparing HR values estimated by the proposed system and the reference one, considering all the volunteers for a total number of 149 beats. The mean absolute percentage error (i.e., MAPE^HR^) was also calculated as follows:(3)$$MAPE^{HR}=\frac{1}{n}\cdot \sum \frac{\left|HR^{smart\_textile}-HR^{reference}\right|}{HR^{reference}}\cdot 100$$
where $HR^{smart\_textile}$ and $HR^{reference}$, values estimated by the smart textile and the PPG sensors, respectively. Results of the beat-by-beat analysis are shown in Figure 8 and Table 3. The good agreement between the f_R_ values measured by the proposed system and the reference one is confirmed by the high value of the R^2^ for both FBG^T^ and FBG^A^ and by the low value of both MOD and MAPE^HR^.

## 4. Tests on Archers During Shooting Sessions

### 4.1. Population and Experimental Protocol

The ST was tested on two archers (a male and a female). Their characteristics are listed in Table 4.

Archers were invited to perform two shooting sessions. Each session consists of six arrows to be shot in five minutes at designed 70 m targets. The first session is a practice round. In this round the arrows are shot at the beginning and do not count as part of the score. Instead, the second one is a scoring round with an awardable maximum score of 60 points (10 points per arrow).

### 4.2. Experimental Set-Up

During each session of shooting, archers worn the flexible sensors on the same positions investigated during tests on volunteers (see Figure 9). The optical spectrum interrogator (si425, Micro Optics Inc. Hackettstown, NJ, USA) for the acquisition of the FBGs output was placed at 3 m of distance from the archer. The sampling frequency was 250 Hz. Since the assessment of the ST was performed on volunteers (Section 3), no reference instruments were used during the shooting action to not impair the archer movements.

### 4.3. Data Analysis and Results

Changes of FBG^T^ and FBG^A^ output of the two practice shooting sessions are plotted in Figure 10.

The analysis was performed by selecting FBG^T^ output because the respiratory acts and the shooting phases are clearly discernible while FBG^A^ output has a more irregular trend (see Figure 10). FBG^T^ output changes were analyzed following three main phases: *i)* the six shooting actions were selected, as shown in Figure 10; *ii)* for each shooting action, the signal related to the aiming phase was filtered and its minimum peaks were detected to calculate the HR values (Figure 11); *iii**)* the signal related to the breathing activity which precedes the shooting actions was filtered and its maximum peaks were detected to estimate the f_R_ values. Results in terms of f_R_ and HR for all the four trials are summarized in Table 5.

## 5. Discussion and Conclusions

This work is focused on cardio-respiratory monitoring in archery using a custom ST based on flexible FBGs.

The feasibility of the system was assessed on nine volunteers during both quiet breathing and apnea. The system showed promising results in terms of both f_R_ and HR estimation, as shown in Table 2 and Table 3. The position of FBGs does not influence the system performance in the f_R_ estimation, while FBG^T^ allows a more accurate HR monitoring than FBG^A^ (see Table 2 and Table 3).

In a previous work [19], we already characterized the proposed sensing element. The flexible FBG sensors showed sensitivity to strain of 0.125 nm⋅mε^−1^, sensitivity to temperature changes of 0.012 nm⋅°C^−1^ and negligible influence of relative humidity on its response. We were the first group that used Dragon skin® 20 as polymer matrix to improve the sensor robustness and skin adherence for the cardio-respiratory monitoring. The main novelties of this work were the assessment of the custom ST on volunteers for the monitoring of both respiratory and cardiac activities and the ST application on archers during shooting sessions.

In the literature, FBGs encapsulated into flexible materials for cardio-respiratory monitoring were mainly proposed for clinical applications (e.g., during magnetic resonance exams). These technological solutions consist of FBG sensors encapsulated into polydimethylsiloxane, PDMS [27], polyvinyl chloride, PVC [28], and fiberglass [21]. In [27], an FBG was encapsulated into PDMS matrix (dimensions: 85 mm × 85 mm × 5 mm). Tests were carried out by positioning the sensing element on the back of two males and two females during MR examination. Results showed maximum relative errors of 4.41% and 5.86% for respiratory and cardiac monitoring, respectively. In [28], PVC was used as flexible matrix to house an FBG for simultaneous respiratory and cardiac monitoring. System showed good performances in the estimation of f_R_ and HR. In [21], an FBG was housed into fiberglass matrix (dimensions: 30 mm × 10 mm × 0.8 mm). The accuracy of the proposed sensor was characterized by relative error <4.64% for f_R_ and <4.87% for HR. In the present study, tests performed on volunteers showed promising results in both the estimation of f_R_ and HR (i.e., MAPE^f^^R^ ≤ 1.97% and MAPE^HR^ ≤ 5.74%).

Focusing on the shooting sessions, results showed that the proposed ST can detect all the six shooting actions and, in turn, monitor f_R_ during the breathing activity and HR during the aiming phase. In the literature, commercially available devices able to monitor only one of these parameters (f_R_ and HR) were used on archers. Breathing activity and patterns were studied in [9] by using Zephyr Bio-Harness devices (Model PSM Research version 1.5, single transmitter and receiver, Annapolis, MD, USA). In [2,5,7] HR values were monitored by using Polar FT4, three silver-silver chloride chest electrodes, and Fitbit Charge HR (Fitbit, Inc. Boston, MA, USA), respectively.

These studies showed that both f_R_ and HR are important since they influence archers’ performance. Therefore, the main improvement of the proposed system is the possibility to monitor both f_R_ and HR with high accuracy during the different phases of the shooting action using the same sensing element. Since the sensing element can be connected to the optical interrogator by means of long, flexible and lightweight fiber optic, the proposed system does not impair the shooting action. This feature encourages further assessment of the proposed system in sports science applications (e.g., during walking running on treadmill and during cycling).

In future works, a high number of archers will be enrolled to investigate how f_R_ and HR influence the shooting performance, and how HR can be estimated in the presence of breathing. These findings will aid the optimization of training strategies according to the experience of each archer and the maximization of their shooting action and scores.

## Figures and Tables

**Figure 1 sensors-19-03581-f001:**
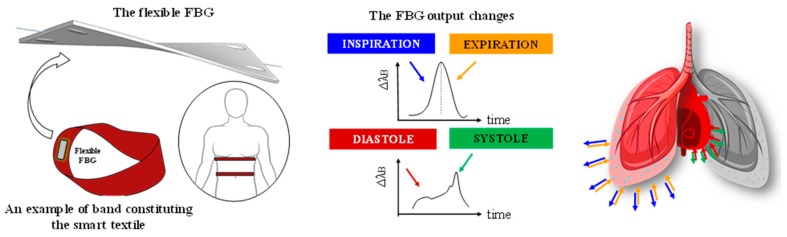
Schematic representation of the sensing element and the smart textile with typical FBG output changes induced by breathing and heart beating.

**Figure 2 sensors-19-03581-f002:**
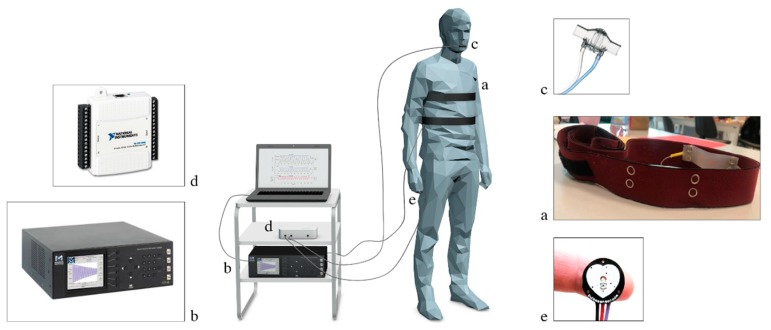
Experimental set-up: (**a**) smart textile consisting of two elastic bands instrumented by flexible FBGs, (**b**) FBG interrogator, (**c**) flow sensor, (**d**) DAQ, and (**e**) PPG sensor.

**Figure 3 sensors-19-03581-f003:**
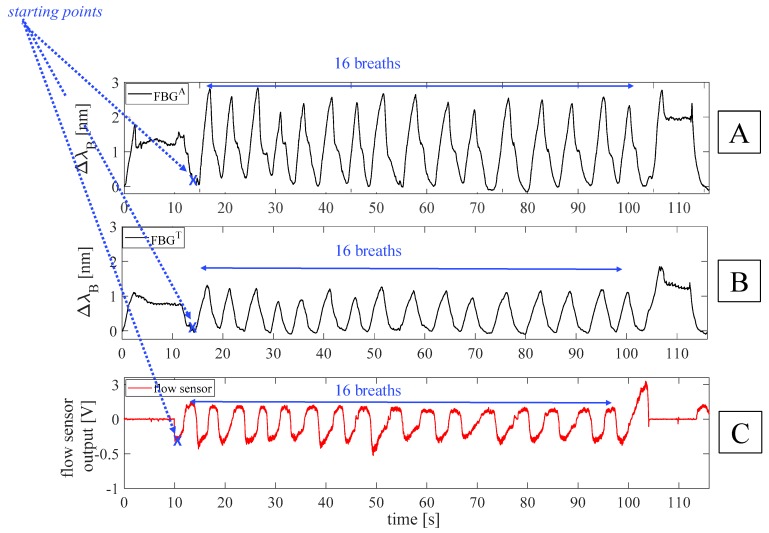
Outputs of the FBGs embedded into the smart textile and of the flow sensor on a whole experiment performed by a volunteer: (**A**) output changes of FBG^A^, placed in correspondence of the umbilicus; (**B**) output changes of FBG^T^, placed in correspondence of xiphoid process; (**C**) output of the flow sensor.

**Figure 4 sensors-19-03581-f004:**
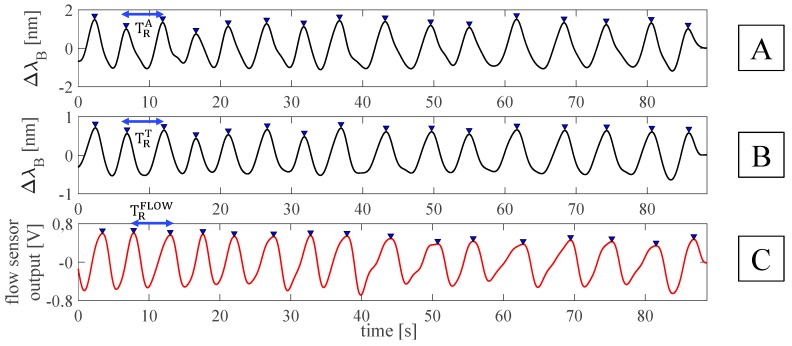
Outputs of the FBGs embedded into the smart textile and of the flow sensor during quiet breathing: (**A**) filtered output changes of FBG^A^; (**B**) filtered output changes of FBG^T^; (**C**) filtered output changes of the flow sensor. The respiratory periods for the band around the abdomen (i.e., T_R_^A^), around the thorax (i.e., T_R_^T^), and for the flow sensor (i.e., T_R_^FLOW^) were also reported.

**Figure 5 sensors-19-03581-f005:**
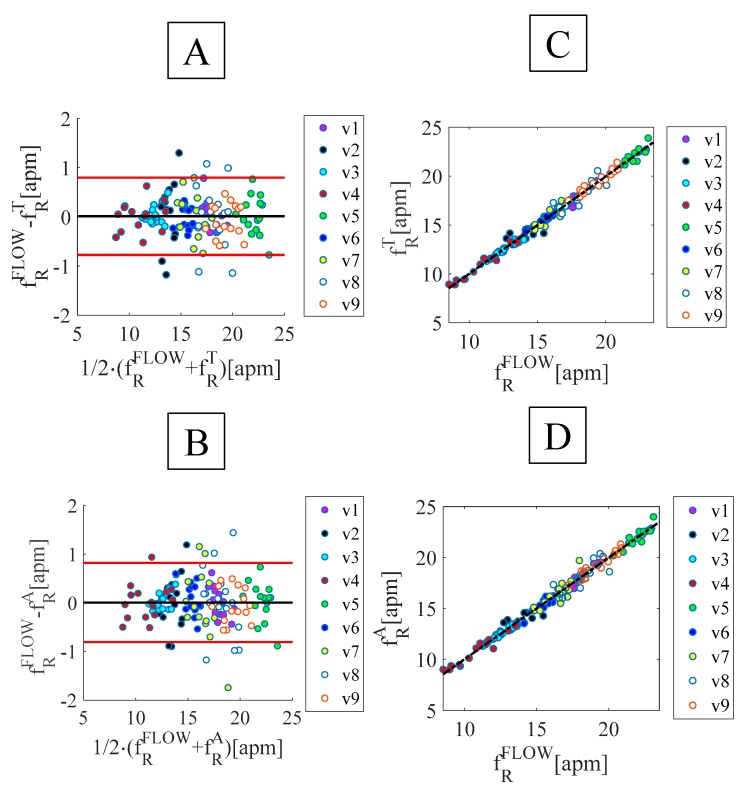
Breath-by-breath analysis: (**A**) and (**B**) Bland-Altman plots using the FBGs placed on the umbilicus, FBG^A^ and xiphoid process, FBG^T^, (**C**) and (**D**) linear regression of the results obtained by the FBGs placed on the umbilicus -FBG^A^- and xiphoid process -FBG^T^- vs. the reference system.

**Figure 6 sensors-19-03581-f006:**
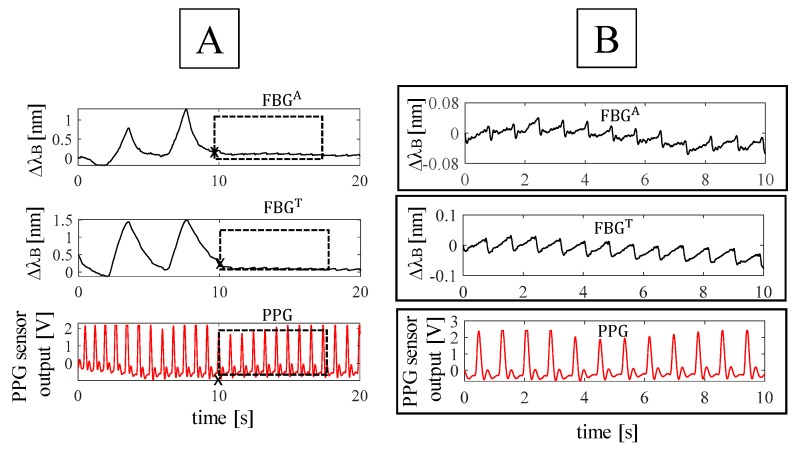
(**A**) Starting points selected on the synchronized signals and (**B**) zoom of the 10 s-window of apnea considering the outputs of FBG^A^, FBG^T^, and PPG.

**Figure 7 sensors-19-03581-f007:**
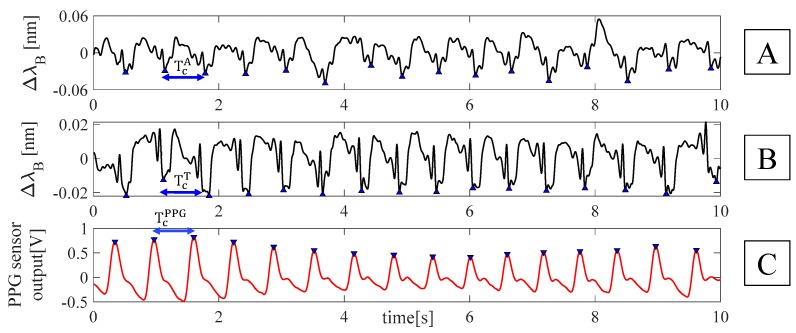
Peaks detection on the filtered signal: (**A**) filtered output changes of FBG^A^, (**B**) filtered output changes of FBG^T^, and (**C**) filtered output changes of PPG sensor. The cardiac periods for the band around the abdomen (i.e., T_C_^A^), around the thorax (i.e., T_C_^T^), and for the flow sensor (i.e., T_C_^PPG^) were also reported.

**Figure 8 sensors-19-03581-f008:**
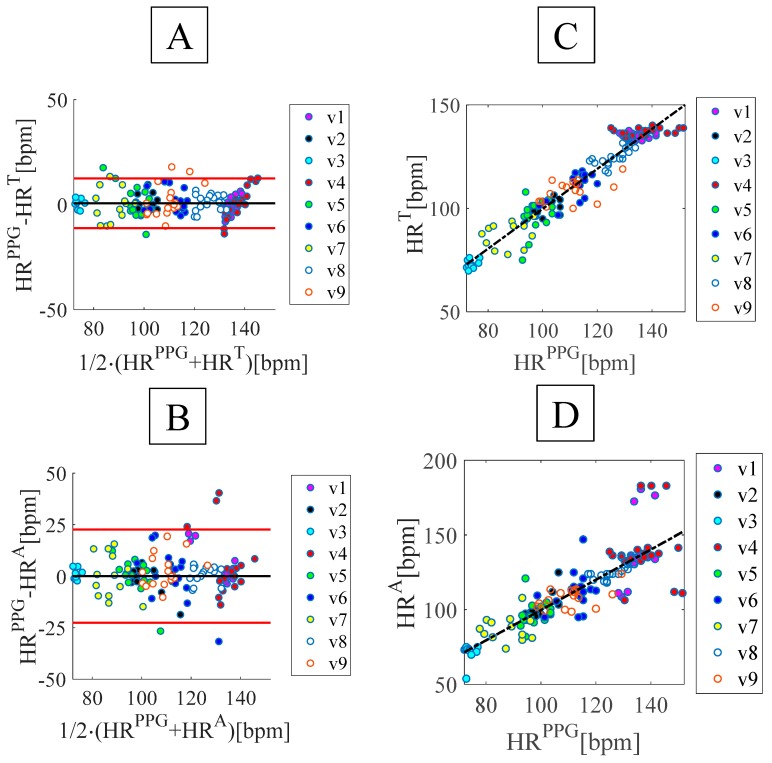
(**A**) and (**B**) Bland-Altman plots using FBG^A^ and FBG^T^, (**C**) and (**D**) linear regression of the results obtained by FBG^A^ and FBG^T^ vs. the reference system.

**Figure 9 sensors-19-03581-f009:**
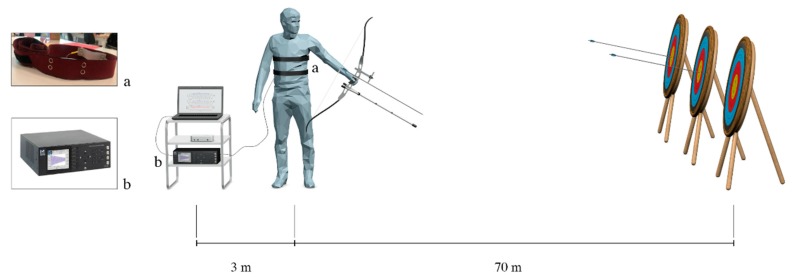
Experimental scenario during shooting session: (**a**) one of the band worn by the archer at designed 70 m targets and (**b**) the FBG interrogator at 3 m of distance from the archer.

**Figure 10 sensors-19-03581-f010:**
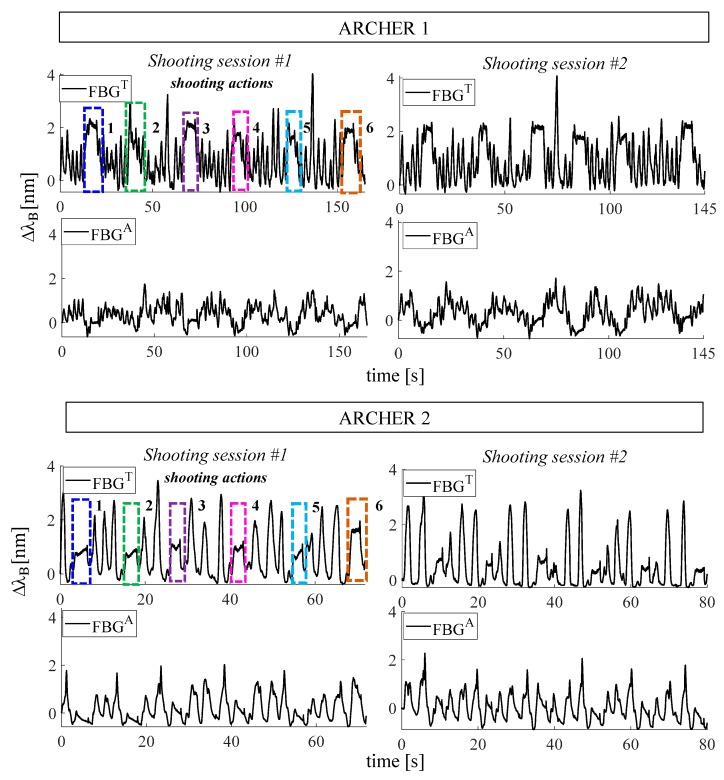
Outputs of the two flexible FBG sensors are shown for the four trials (2 shooting sessions for each archer). By way of example, the shooting actions of the first session are highlighted by dashed rectangular boxes.

**Figure 11 sensors-19-03581-f011:**
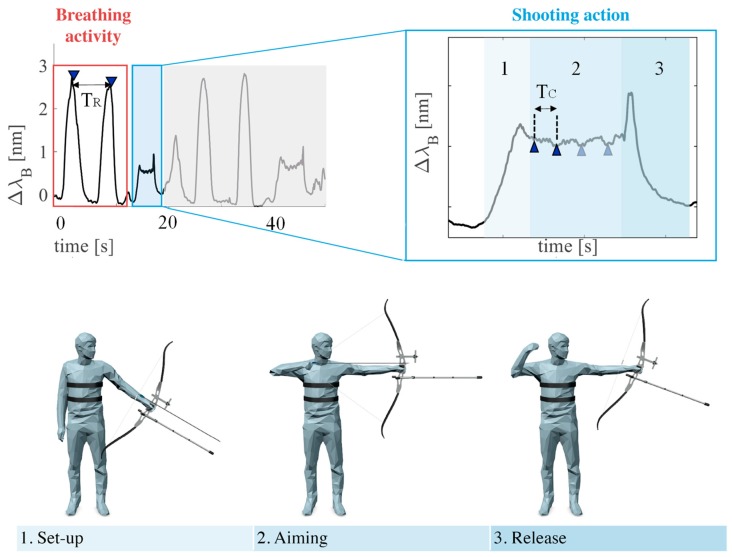
Output of the sensor changes during both breathing activity and shooting action.

**Table 1 sensors-19-03581-t001:** Population.

Volunteer	Age (years)	Height (cm)	Weight (kg)	C_T_^1^ (cm)	C_A_^1^ (cm)
**1**	28	182	70	82	74
**2**	22	168	74	60	80
**3**	30	163	81	62	84
**4**	29	180	82	69	91
**5**	26	153	69	48	71
**6**	22	166	67	58	76
**7**	27	173	82	71	90
**8**	25	160	74	60	78
**9**	22	172	67	55	72

^1^ C_T_: thoracic circumference; C_A_: abdominal circumference.

**Table 2 sensors-19-03581-t002:** Performance of the smart textile in respiratory monitoring: results of Bland-Altman analysis, linear regression and MAPE^fR^.

	R^2^	MOD (apm)	LOAs (apm)	MAPE^fR^ (%)
**f_R_^T^**	0.99	0.014	−0.804; 0.832	1.92
**f_R_^A^**	0.98	0.004	−0.811; 0.819	1.97

**Table 3 sensors-19-03581-t003:** Bland-Altman of beat-by-beat analysis.

	R^2^	MOD (bpm)	LOAs (bpm)	MAPE^HR^ (%)
**HR^A^**	0.76	0.059	−22.54; +22.65	5.74
**HR^T^**	0.91	0.664	−11.15; +12.48	3.92

**Table 4 sensors-19-03581-t004:** Archers’ characteristics.

	Age (years)	Height (cm)	Weight (kg)	C_T_ (cm)	C_A_ (cm)	Experience (years)	Training Frequency (days per week)
**Archer 1**	20	167	65	97	80	3	3
**Archer 2**	33	165	64	75	60	2	6

^1^ C_T_: thoracic circumference; C_A_: abdominal circumference.

**Table 5 sensors-19-03581-t005:** Values of f_R_ and HR during the first and the second shooting sessions.

	Shooting Session #1						Shooting Session #2					
**Shot**	1	2	3	4	5	6	1	2	3	4	5	6
	**f_R_ (apm)**						**f_R_ (apm)**					
**Archer 1**	23.9	24.6	20.0	26.5	21.9	20.5	24.1	25.0	20.6	26.5	25.6	23.9
**Archer 2**	15.2	13.3	9.3	8.6	10.1	8.4	7.4	8.5	7.6	8.8	9.0	7.1
	**HR (bpm)**						**HR (bpm)**					
**Archer 1**	101.8	101.3	97.3	96.7	100.0	94.9	116.4	113.7	108.9	104.7	120.5	115.9
**Archer 2**	87.6	89.2	94.3	89.3	97.3	94.1	97.2	88.3	92.0	92.3	82.1	90.5

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
