# Peer review of "Cardio-Respiratory Monitoring in Archery Using a Smart Textile Based on Flexible Fiber Bragg Grating Sensors"

_sensors, 2019, doi:10.3390/s19163581_

Round 1

Reviewer 1 Report

The authors present the use of a smart textile based on FBGs to monitor heart rate and respiration rate in archery.

The presented topic is interesting and quite original. The manuscript is well organized and dense of experimental evaluations, including tests on both healthy volunteers and archers. Also English language and style are fine.

I only suggest some minor revisions to better clarify some results (see the attached annotated pdf).

Author Response

Primarily we would like to thank the Reviewer for his (her) observations, which helped us to improve the earlier version of the paper.

We appreciate the reviewer taking the time to thoughtfully consider our manuscript. It is always extremely helpful to have someone outside of the research team who can identify sections/parts that need clarification. We are appreciative that the reviewer #1 feels the manuscript almost ready for publication and finds the manuscript interesting.

The provided suggestions have been helpful in order to amend the manuscript, as we hope it emerges from the examination of the new version of the article.

We have changed all the parts according to the recommendation of the  reviewer.

Thanks a lot

Daniela Lo Presti

Reviewer 2 Report

This work contributes to the study to monitor cardio-respiratory activity on archery during shooting using smart textiles. This article is interesting for monitoring vital signs of athletes. I recommend this article will be accepted after the authors consider the comments below.

In the title, I suggest the abbreviation “FBG” is replaced with the “fiber Bragg grating” because there is no explanation in abstract. What does it mean the elastic band with 10 kg of resistance? I suspect that the weight of the band is 10 kg or the band can be stretched under 10kg force. Is it possible to monitor breathing and respiratory rates simultaneously by using this FBG? In this study, the authors evaluate these rates separately. In this experimental study, an optical spectrum interrogator is used for the acquisition of the FBG output. I wonder the system may be obstructive for the archers during shooting. Is there no effect on the actions of archers? Is it possible to apply this system for other sports activity as well as archery? (for example, a runner on treadmill)

Author Response

Response to Reviewer 2

Primarily we would like to thank the Reviewer for his (her) observations, which helped us to improve the earlier version of the paper. 

We appreciate the reviewer taking the time to thoughtfully consider our manuscript. It is always extremely helpful to have someone outside of the research team who can identify sections/parts that need clarification. We are appreciative that the reviewer #2 finds the manuscript the manuscript interesting.

The provided suggestions have been helpful in order to amend the manuscript, as we hope it emerges from the examination of the new version of the article. 

We have changed all the parts according to the recommendation of the reviewer.

GENERAL COMMENTS

This work contributes to the study to monitor cardio-respiratory activity on archery during shooting using smart textiles. This article is interesting for monitoring vital signs of athletes. I recommend this article will be accepted after the authors consider the comments below.

SPECIFIC COMMENTS

Comment #1.In the title, I suggest the abbreviation “FBG” is replaced with the “fiber Bragg grating” because there is no explanation in abstract.

Answer #1.Thanks a lot for the observation, we replaced FBG with fiber Bragg grating.

 Comment #2.What does it mean the elastic band with 10 kg of resistance? I suspect that the weight of the band is 10 kg or the band can be stretched under 10kg force..

Answer #2.Thanks a lot for the observation, we are sorry for the mistake. We replaced the previous expression with “10 kgf of maximum load”

Comment #3.Is it possible to monitor breathing and respiratory rates simultaneously by using this FBG? In this study, the authors evaluate these rates separately.

Answer #3.Thanks a lot for the observation which allows us to clarify this part of the manuscript. Since, breathing and respiratory rates can be considered as synonym, we think the referee means “cardiac and respiratory rates”. As stated by the referee we assessed the performance of the proposed system in terms of respiratory rate and heart rate, separately. At this stage, we assessed the performance of the system in terms of heart rate estimation only during apnea. This facilitates the detection of chest wall motion caused by heart pumping since the much bigger movements caused by breathing are absent. Considering the specific application this is not a real limit, since each archer during the aiming phase of the shooting holds the breath. In order to highlight this aspect, we replaced the final part of the section Discussion and Conclusion:

“Therefore, the main improvement of the proposed system is the simultaneous monitoring of both fRand HR with high accuracy using the same sensing element and without impairing the shooting actions.

In future works, a high number of archers will be enrolled to investigate how fRand HR influence the shooting performance. These findings will aid in the optimization of training strategies according to the experience of each archer and the maximization of their shooting action and scores.”

  With the following:

“Therefore, the main improvement of the proposed system is the possibility to monitor both fRand HR with high accuracy during the different phases of the shooting action using the same sensing element. Since the sensing elements can be connected to the optical interrogator by means of long, flexible and lightweight fiber optic, the proposed system does not impair the shooting action. This feature encourages to test the proposed system in further sports science applications (e.g., during walking running on treadmill and during cycling).   

In future works, a high number of archers will be enrolled to investigate how fRand HR influence the shooting performance and how estimate HR in presence of breathing. These findings will aid in the optimization of training strategies according to the experience of each archer and the maximization of their shooting action and scores.”   

Comment #4.In this experimental study, an optical spectrum interrogator is used for the acquisition of the FBG output. I wonder the system may be obstructive for the archers during shooting. Is there no effect on the actions of archers? Is it possible to apply this system for other sports activity as well as archery? (for example, a runner on treadmill).

Answer #4.Thanks a lot for the observation, we highlighted these aspects at the end of the section discussion and conclusion. The new part is the following:

“Therefore, the main improvement of the proposed system is the possibility to monitor both fRand HR with high accuracy during the different phases of the shooting action using the same sensing element. Since the sensing element can be connected to the optical interrogator by means of long, flexible and lightweight fiber optic, the proposed system does not impair the shooting action. This feature encourages to test the proposed system in further sports science applications (e.g., during walking running on treadmill and during cycling).   

In future works, a high number of archers will be enrolled to investigate how fRand HR influence the shooting performance and how estimate HR in presence of breathing. These findings will aid in the optimization of training strategies according to the experience of each archer and the maximization of their shooting action and scores.”

Thanks a lot